# STARD3: A New Biomarker in HER2-Positive Breast Cancer

**DOI:** 10.3390/cancers15020362

**Published:** 2023-01-05

**Authors:** Massimo Lodi, Laetitia Voilquin, Fabien Alpy, Sébastien Molière, Nathalie Reix, Carole Mathelin, Marie-Pierrette Chenard, Catherine-Laure Tomasetto

**Affiliations:** 1Department of Pathology, Strasbourg University Hospital, 67200 Strasbourg, France; 2Surgical Oncology Department, Institut de cancérologie Strasbourg Europe (ICANS), 17 Rue Albert Calmette, 67200 Strasbourg, France; 3Institut de Génétique et de Biologie Moléculaire et Cellulaire (IGBMC), CNRS UMR7104, INSERM U1258, Université de Strasbourg, 1 Rue Laurent Fries, 67400 Illkirch-Graffenstaden, France

**Keywords:** breast cancer, human epidermal growth-factor receptor 2 (HER2), neo-adjuvant systemic treatment, STARD3, pathological complete response

## Abstract

**Simple Summary:**

One out of 8 breast cancers is of a particular type called “HER2-positive”, which is more aggressive and is treated by surgery and a medical treatment associating chemotherapy and a targeted drug against the protein HER2, possibly followed by radiotherapy and endocrine therapy. When the medical treatment is performed before others (neoadjuvant) and that all cancer cells are gone, this is called “pathological complete response”, which is an important good-prognosis marker. This article studied on a group of 112 patients with HER2-positive breast cancer, the presence of a protein called STARD3, which has a strong biological and genetic association with HER2. It found that STARD3 was strongly associated with pathological complete response and could predict this response. It also investigated if the presence of STARD3 had an impact on prognosis (such as survival and the risk of cancer recurrence). This work suggests that the study of STARD3 for HER2-positive breast cancers could lead to a better treatment planning, and further studies are needed.

**Abstract:**

Pathological complete response (pCR) after neoadjuvant systemic treatment (NST) is an important prognostic factor in HER2-positive breast cancer. The majority of HER2-positive breast cancers are amplified at the HER2 gene locus, several genes are co-amplified with HER2, and a subset of them are co-expressed. The STARD3 gene belongs to the HER2 amplicon, and its role as a predictive marker was never addressed. The objective of this study was to investigate the predictive value of STARD3 protein expression on NST pathological response in HER2-positive breast cancer. In addition, we studied the prognostic value of this marker. Methods. We conducted a retrospective study between 2007 and 2020 on 112 patients with non-metastatic HER2-positive breast cancer treated by NST and then by surgery. We developed an immunohistochemistry assay for STARD3 expression and subcellular localization and determined a score for STARD3-positivity. As STARD3 is an endosomal protein, its expression was considered positive if the intracellular signal pattern was granular. Results: In this series, pCR was achieved in half of the patients. STARD3 was positive in 86.6% of cases and was significantly associated with pCR in univariate analysis (*p* = 0.013) and after adjustment on other known pathological parameters (*p* = 0.044). Performances on pCR prediction showed high sensitivity (96%) and negative predictive value (87%), while specificity was 23% and positive predictive value was 56%. Overall, specific, relapse-free, and distant metastasis-free survivals were similar among STARD3 positive and negative groups, independently of other prognosis factors. Conclusion: NST is an opportunity for HER2-positive cancers. In this series of over a hundred HER2-positive and non-metastatic patients, a STARD3-negative score was associated with the absence of pathological complete response. This study suggests that determining STARD3 overexpression status on initial biopsies of HER2-positive tumors is an added value for the management of a subset of patients with high probability of no pathological response.

## 1. Introduction

Breast cancer is a heterogeneous disease, composed of different biological subtypes which have distinct behaviors and prognosis. Amplification of the human epidermal growth factor receptor 2 (HER2) is present in 13–15% of cases [1] and leads to a biologically aggressive malignancy with increased sensitivity to chemotherapy. Today, in the clinical practice, HER2 positivity is established by pathologists using immunohistochemistry (IHC), possibly in situ hybridization (ISH) and follows international guidelines [2]. HER2-positive cancers have a specific treatment sequence based on the combination of chemotherapy and anti-HER2 targeted therapies. This specific patient care protocol includes, in most cases, a systemic treatment of cytotoxic chemotherapy (with or without anthracyclines), which is associated with targeted anti-HER2 therapies blocking either the receptor itself or its tyrosine kinase activity. Over the years, clinicians have noted that in the case of non-metastatic diseases, neoadjuvant systemic treatment (NST) has multiple benefits. First, NST reduces tumor size and favors breast-conserving surgery with a better aesthetic result. Second, it accelerates systemic treatment, which is beneficial in case of the presence of occult distant disease and when surgery must be delayed. This might happen during pregnancy or during oncogenetic testing, given that the results can change the surgical treatment. Third, NST allows for the evaluation of tumoral response to a given therapeutic regimen. Notably, pathological response to NST gives a direct indication of treatment efficiency and constitutes an important prognostic factor. The presence of residual invasive disease either in the breast or in axillary nodes after NST with single or dual HER2-targeted therapy is an indication for a postoperative adjuvant treatment with the next generation of anti-HER2 therapy ado-trastuzumab emtansine (T-DM1). The number of anti-HER2 therapies is continuously increasing, and with the expansion of this therapeutic arsenal, there is a need to properly tailor anti-HER2 targeted treatment. Moreover, given that these treatments are cost-effective and may have deleterious side-effects, much attention is being paid to improve the prediction of clinical benefits. Some clinicopathological factors are associated with pCR such as high grade, hormone receptor negativity, absence of lymphovascular invasion, or high Ki67 [3,4,5]. Conversely, large tumor size [6] and lymphovascular invasions [7] are inversely associated with pCR. Compared with the other subtypes, HER2+ breast cancers achieve higher rates of pCR [5], with the rates that vary between clinical trials being estimated to reach 50% with taxanes-anthracyclines-based regimens in association with trastuzumab [8,9,10]. Other regimens without anthracyclines have been published, such as in the TRAIN study comparing carboplatin-taxanes-trastuzumab with or without pertuzumab (pCR rates of 67 versus 68%, respectively, not significantly different] [11,12]) or in the TRYPHAENA study comparing anthracyclines, docetaxel, carboplatin, trastuzumab, and pertuzumab [13]. Nonetheless, a number of cases do not respond to NST, and there is a need to find additional predictive factors. In our laboratory, we have identified a gene named StAR-related lipid transfer domain-3 (STARD3) co-amplified and co-expressed with HER2 in breast cancer [14,15]. STARD3 is present in the HER2 minimal region of amplification in cancers [16,17,18], and importantly, STARD3 functions in HER2+ cancer cells. The reduction of STARD3 expression in HER2-positive cancer cell lines reduces their growth [19,20,21]. Of interest, Salberg et al. noted that cell lines resistant to anti-HER2 therapies were still dependent on STARD3 expression [21], suggesting that STARD3 has a unique role in the biology of HER2-positive cancer cells and may be a predictive factor for pCR.

The objective of this study was to investigate the prognostic and predictive value of STARD3 protein expression on NST pathological responses in HER2-positive breast cancer.

## 2. Materials and Methods

### 2.1. Expression Analyses from Public Databases

Data were extracted from the Cancer Cell Line Encyclopedia website, Broad Institute of MIT and Harvard [22]. We collected mRNA expression (RNAseq) and DNA copy number for each gene on 51 different breast cancer cell lines (Appendix A). Statistical analyses and graphical plotting were carried out using R version 4.1.3 (10 March 2022) [23] and R packages *Hmisc*, *ggpubr*, *ggplot2*, and *GEOquery.* For correlation, we used Pearson correlation test, and the regression line was made with a linear model. A breast invasive carcinoma study was identified in cBioPortal (cbioportal.org) for Cancer Genomics [24]. Genomic profiles corresponding to mRNA expression z score relative to all samples (log microarray) were selected. The ERBB2 and STARD3 genes were queried. The correlation between ERBB2 and STARD3 mRNA expression was determined using the plot tab, and horizontal axis and vertical axes corresponded to STARD3 and ERBB2 mRNA expression (microarray), respectively. The regression line tab was selected. The Gene Expression Omnibus (GEO) repository [25] was queried for HER2-positive breast cancers specimens treated with similar neoadjuvant systemic treatment (see eligibility criteria) and measuring pathological complete response. Two studies were found, evaluating gene expression by RNA sequencing, including *n* = 24 [26] and *n* = 57 specimens [27]. A comparison of STARD3 RNA between the “pCR” and “no pCR” groups was performed with a Wilcoxon test.

### 2.2. Cell Culture and Transfections

HCC1954 (American Type Culture Collection CRL-2338) cells were maintained in RPMI w/o HEPES with 10% fetal calf serum (FCS) and 40 μg/mL gentamycin. HeLa (ATCC CCL-2) cells were maintained in DMEM with 5% FCS and 40 μg/mL gentamycin. Transient siRNA transfections were performed using Lipofectamine RNAiMAX (Invitrogen, P/N 56532) according to the manufacturer’s instructions. Control siRNA (D-001810-10) and STARD3-targeting siRNAs (L-017665-00) were from SMARTpool ON-TARGETplus obtained from Horizon Discovery. siRNAs were used at 10 nM final concentration and cells were transfected 24–72 h prior to experiments. Plasmid transient transfections were done in 6-well plate using X-tremeGENE 9 DNA Transfection Reagent (Roche) and 1 μg of pEGFP-STARD1, pEGFP-STARD2, pEGFP-STARD3, pEGFP-STARD4, pEGFP-STARD5, or pEGFP-STARD6. Protein extracts were obtained 48 h after transfection by scraping phosphate-buffered saline 1X (PBS) washed cells with 300 μL of lysis buffer (150 mM Tris.HCl, Ph 7.4, 150 mM NaCl, 1 mM EDTA, 1% Triton X-100, and 1X Complete protease inhibitor from Roche). After 15 min on ice, cell extracts were centrifuged for 10 min at 10,000 rpm, and supernatants were collected and quantified.

### 2.3. Immunofluorescence and Western Blot

HCC1954 cells were grown to 70% confluence on glass coverslips. After washing with PBS 1X, cells were fixed 10 min at room temperature in 4% paraformaldehyde in PBS 1X then permeabilized for 10 min with 0.1% Triton X-100 in PBS. After blocking in 1% bovine serum albumin in PBS, cells were incubated at 4 °C overnight with the primary mouse anti-STARD3 3G11 (1:1000) antibody (IGBMC). Secondary antibody (1:1000) was Donkey anti Mouse Alexa-555 (ThermoFisher Scientific, Waltham, MA, USA). Nuclei were counterstained with the Hoechst-33258 dye (ThermoFisher Scientific, Waltham, MA, USA). Slides were mounted in ProLong Gold (Invitrogen, Waltham, MA, USA). Observations were made with a confocal microscope (Leica SP8 UV, 63×, NA 1.4). For Western blot analysis, nearly equal amounts of proteins (20 μg) were separated into 7–14% SDS–PAGE and transferred onto nitrocellulose membranes. Membranes were blocked with non-fat dry milk 3% in 1× PBS, Tween-20 0.1%, and incubated overnight at 4 °C with anti-STARD3 (3G11; 1/1000), anti-GFP (1/2000) (TP401, Torrey Pines Biolabs, Secaucus, NJ, USA), and anti-Rab7 (#2576 1/1000). Secondary horseradish peroxidase (HRP) conjugated anti-Mouse and anti-Rabbit antibodies were from Jackson ImmunoResearch. Signals were acquired using the LAS Quant 4000 System (GE Healthcare, Uppsala, Sweden). All the whole western blot figures can be found in Appendix A.

### 2.4. STARD3 Immunohistochemistry Assay on Human Breast Cancer

#### 2.4.1. Study Design

We conducted a retrospective analysis of STARD3 expression in HER2-positive human breast cancers. For that, we constituted a cohort of patients diagnosed with a HER2-positive primary breast cancer by routine immunohistochemistry (score 3+) treated at the Strasbourg University Hospital between 2007 and 2020. This clinical study included 112 HER2-positive breast cancer patients treated with NST at Strasbourg University Hospitals and was conducted following the Strengthening the Reporting of Observational Studies in Epidemiology (STROBE) guidelines [28].

#### 2.4.2. Eligibility Criteria

Inclusion criteria were: (1) primary breast cancer with stage I to III at diagnosis; (2) HER2-positive breast cancer defined by a IHC score of 3+; (3) surgical treatment and pathological analysis of the operative specimen at the Strasbourg University Hospitals; (4) anthacycline-based NST with trastuzumab [(F)EC-TH protocols]: epirubicin and cyclophosphamide with or without fluorouracil (respectively, FEC and EC) every three weeks for three to four cycles, followed by taxanes (weekly paclitaxel for nine to twelve cycles or docetaxel every three weeks for three to four cycles) and trastuzumab every three weeks; and 5) patient informed consent. Patients with initial metastatic disease and those for whom the initial core biopsy was not available for analysis were excluded. Patient selection was resumed in the CONSORT diagram (see Figure 1).

#### 2.4.3. Study Endpoints

Primary endpoint was the detection and location of STARD3 protein expression by IHC in core biopsies before NST and its performances in pCR prediction. The secondary endpoint was survival (overall, specific, and event-free) according to STARD3 expression.

#### 2.4.4. Patient Selection

We performed an initial query in the pathology laboratory database of the Strasbourg University Hospital (DIAMIC^®^, INFOLOGIC-Santé, Valence, France). Then, we gathered different clinical and pathological parameters for each patient from the electronic health record. According to the therapeutic response, we divided the cohort into two groups: “pCR” and “no pCR”. pCR was defined by a complete tumor regression after NST, and included patients with no residual cancer cells (ypT0N0 according to the TNM classification [29]) or with residual in situ neoplasia and the absence of infiltrative cancer (ypTisN0). The “no pCR” group included all the other patients.

#### 2.4.5. Clinical and Pathological Characteristics

Hormone receptor and HER2 expression were quantified according to the American Society of Clinical Oncology (ASCO)/College of American Pathologists (CAP) guidelines [2,30]. Tumors were considered hormone receptor positive if H-Score was >10. HER2 was considered negative if the score was 0 or 1+ and positive for 3+. In the case of HER2 score 2+, silver in situ hybridization (SISH) was performed to assess HER2 amplification. Nuclear grade was described according to the Elston and Ellis modified Scarff–Bloom–Richardson (SBR) grading system [31]. Tumor stage was defined according to the TNM Classification of the American Joint Committee on Cancer classification [29]. Survival was calculated from time of diagnosis. We considered a patient lost to follow-up if the last known date was greater than 1 year and the patient did not complete the recommended follow-up program.

#### 2.4.6. STARD3 Immunohistochemistry Protocol

For each patient, we gathered clinical and pathological data. Then, we unarchived the initial core biopsy of the tumor, which was previously fixed in formaldehyde and embedded in a paraffin block. Each block was cut in 4 μm slices using a microtome that were placed on positive-charged glass slides [VWR SuperFrost^®^ Plus, Rosny-sous-Bois, France] after a bath in 40 °C water. Each slide was identified with a bar code and specimen and protocol identification. We then proceeded to automated deparaffinization and staining using BenchMark Ultra IHC slide staining system automat (Ventana, Roche, Basel, Switzerland) with UltraView Universal DAB detection kit (Ventana, Roche). The first step was deparaffinization by heating slides to 72 °C then washing them four times with EZ Prep™ 1x (950-102, Ventana, Roche) solution. Then, a pre-treatment was performed by association of heating to 95 °C and Cell Conditioning 1™ solution (CC1, Ventana, Roche) with the 64-min standard protocol. At the end of pre-treatment, slides were at 36 °C until the end of the whole protocol. Slides were washed 3 times with reaction buffer. Endogenous enzyme inhibition was performed with UltraView Inhibitor™ (Ventana, Roche) applied for 4 min. Slides were washed 2 times with reaction buffer. Then, anti-STARD3 3G11 primary antibody (mouse monoclonal antibody, IGBMC, [32]) at 1/500 dilution was applied for 32 min at 36 °C. Initially, different anti-STARD3 antibodies and different dilutions were tested in order to choose the best experimental situation. Slides were washed two times with reaction buffer. Next, secondary anti-mouse antibody coupled with horseradish peroxidase (UltraView HRP Multimer™ (Ventana, Roche)) was applied for 8 min at 36 °C. Slides were washed 3 times with reaction buffer. At this point, UltraView DAB™ and DAB H_2_O_2_™ (Ventana, Roche) were applied on the slides for 8 min at 36 °C, and UltraView Copper™ (Ventana, Roche) was used to improve staining readability. Slides were washed once with reaction buffer. Ultimately, a counterstaining with Hematoxylin II (Ventana, Roche) for 8 min was performed. Slides were dehydrated with ethanol and xylene, then mounted with the machine Cover Tech Microm CTM6 (Thermo Scientific) using Pertex^®^ rapid drying medium for mounting (Histolab, Askim, Sweden) and 24 × 50 mm cover slips (Mezel-Glöser, VWR). For each patient, we examined STARD3 and HER2 immunohistochemistry assays using brightfield microscopy, and those slides were scanned with NanoZoomer-XR (Hamamatsu, Shizuoka, Japan) using the whole slide imaging system.

#### 2.4.7. Statistical Analysis

Statistical analysis was performed with R version 4.1.3 (10 March 2022) [23]. When comparing qualitative variables, we performed two-sided Chi^2^ tests with Yates’s correction (or Fisher’s exact tests if the samples sizes were small). When comparing quantitative variables (such as age) we used two-sided Wilcoxon tests. We measured the test performance using sensitivity, specificity, positive and negative predictive value, and calculated their 95% confidence intervals (95% CI). For multivariate analysis of dichotomous variables (pCR versus no pCR), we performed a binomial logistic regression with a generalized linear model (*logit* model). Odds ratio and their 95% CI were obtained with exponential of each variable coefficient and confidence intervals. For survival analysis, we used the Kaplan–Meier method and the log-rank test, and the Cox Proportional Hazard survival models for multivariate analysis.

## 3. Results

### 3.1. A Strong Association between STARD3 and HER2 in Breast Cancers

Gene amplification is an important mechanism in cancer, leading to increased gene expression and gain of function [33]. In our previous studies, using cell lines and breast tumors, we showed that STARD3 gene amplification and overexpression paralleled that of HER2 [14,15,34]. In order to precisely establish the relationship between DNA copy number and RNA expression levels for STARD3 and other genes from the HER2 amplicon in independent samples, we took advantage of the CCLE (Cancer Cell Line Encyclopedia) project [22]. We extracted DNA copy number and RNA expression data for HER2, GRB7, STARD3, and TOPA2 genes in 51 breast cancer cell lines (Appendix A). The correlation coefficient between DNA copy number and RNA expression was calculated for the different genes tested. STARD3 expression showed a strong positive correlation with its own DNA copy number (Figure 2A), a feature shared by HER2 and Growth factor receptor-bound protein 7 (GRB7) (Figure 2B). More important, STARD3 DNA copy number showed a strong positive correlation with HER2 DNA copy number, and both STARD3 DNA copy number and its expression (RNA level) are strongly correlated with that of HER2 (Figure 2). We next explored a large-scale cancer genomics database using the cBioPortal resource [24]. In this collection of invasive breast tumors, STARD3 expression was positively correlated with HER2 expression (Figure 2C).

Several studies reported a poor prognostic value for STARD3 expression in breast cancers [35,36]. However, data about its potential interest as a predictive biomarker regarding therapeutic intervention are still missing. In this study, we investigated STARD3 expression on a cohort of HER2-positive breast cancers. With the aim of examining STARD3 expression by immunohistochemistry (IHC), we tested the specificity of an anti-STARD3 antibody developed in our laboratory [32]. We first analyzed STARD3 expression by western blot on a HER2-positive cell line (HCC1954). The 3G11 antibody detected STARD3 in wild-type cells and cells transfected with control small interfering RNAs (siRNAs) while no signal was detected in cells treated with specific siRNAs against STARD3 (Figure 3A). Next, because STARD3 belongs to a family of 15 related proteins in mammals [37], we studied signal specificity with respect to the other members of the START protein family. To this aim, different constructs expressing fusion proteins between the GFP and a number of STARD proteins were transfected in HeLa cells and analyzed by western blot. We expressed STARD1 and STARD4-D6, as they are the STARD3 closest homologs in the family. The anti-STARD3 antibody recognized only the GFP-STARD3 construct (Figure 3B), indicating its specificity. We then tested the antibody by immunofluorescence on HCC1954 cells. This experiment showed that the anti-STARD3 antibody recognized a punctate cytoplasmic signal typical of STARD3 in wild-type and control cells, while the signal was lost in cells transfected with siRNAs against STARD3 (Figure 3C). Finally, the antibody was tested by IHC on fine needle breast biopsies. As shown in Figure 3D, the antibody detected a specific punctate or granular signal in the cytoplasm of cancer cells, while normal epithelial and stromal cells were negative.

Altogether, these data confirmed the strong correlation between STARD3 and HER2 DNA amplification and RNA expression in breast cancer and prompted us to examine STARD3 protein expression in a clinical series by IHC using a specific anti-STARD3 antibody.

### 3.2. Patient Characteristics

To study the predictive value of STARD3 protein expression regarding therapeutic intervention, we conducted a retrospective clinical study on HER2+ tumors. Tumor samples were obtained between 2007 and 2020 from 112 HER2-positive breast cancer patients treated with NST at Strasbourg University Hospitals (for patient selection, see Figure 1). Table 1 shows the clinical, morphological, and treatment characteristics of the study cohort classified according to pCR. Of the 112 patients included, 56 achieved pCR (50%) and 56 did not (50%). The median age at diagnosis, the presence of multifocal disease, the histological subtype, and the initial tumor size were similar amongst the two groups. Likewise, there were no differences associated with the distinct NST regimens used. Of interest, ER expression was correlated with pCR, as lower ER H-Scores were observed in the pCR group (91.7 versus 178.8, *p* < 0.001). A similar tendency was observed with PR expression, but it was not significant. Of note, high grade (SBR III) was associated with pCR (73.2% versus 48.2%, *p* = 0.007).

These data show that this cohort is representative of HER2+ breast-cancer treated with NST, as pCR rates are similar to those reported in other studies, and clinicopathological factors associated with pCR are consistent with the literature.

### 3.3. Immunohistochemistry on Initial Biopsies

We analyzed by IHC STARD3 protein expression in 112 HER2-positive invasive breast cancers on the initial tumor biopsies. The tumors analyzed had been classified in the HER2-positive subgroup by routine immunohistochemistry performed on the same biopsy. STARD3 expression was scored according to signal intensity, the number of positive cancer cells, and staining patterns (Figure 4 and Table 2). Intensity was evaluated with a 4-value subjective scale from 0 (none) to 3 (very strong) (Figure 4A). High STARD3 expression was found in the majority of samples: out of 112 samples, 104 tumors were positive and 8 were negative (Table 2). We also looked at the percentage of cancer cells positive for STARD3 in these samples. In the 104 STARD3-positive tumors, the protein was overexpressed in almost all cancer cells, but in rare cases, heterogeneity was observed (Table 2). Consistent with the known localization of STARD3 in endosomes [38], in most positive cases (97 out of 104), the STARD3 staining pattern was granular (Figure 4 and Table 2). In the majority of cases, the granular staining was dispersed in the cytosol (Figure 4B(a,c)), and in fewer cases the granular signal was polarized either close to the plasma membrane or in the perinuclear area (Figure 4B(b)). Finally, in some tumors, no signal or a weak diffuse signal was observed (Figure 4A(c) and 4B(d)). For these tumors, the overexpression and amplification of HER2 was controlled (Figure 4A(c)). All these cases had HER2 overexpression (controlled by HER2 score 3+ in immunohistochemistry assays). Silver in situ hybridization (SISH) was performed for all STARD3 negative cases (*n* = 15), and we found that among them 12 had HER2 amplification while 3 were not amplified, suggesting that in these tumors HER2 overexpression is independent from amplification and is caused by transcription. Interestingly, all STARD3-negative and HER2-positive but non-amplified (SISH-negative) cases did not achieve pCR.

Amplification at the HER2 locus is not specific to invasive carcinoma. Many studies reported that ductal carcinoma in situ (DCIS) have HER2 amplification and overexpression [39]. In particular, one study in pure DCIS showed co-amplification and co-expression along with HER2 of several neighboring genes, including STARD3 and GRB7 [40]. Because the presence of DCIS is frequent in invasive carcinoma, we looked for STARD3 expression in the in situ and invasive components of tumors from our series. We found 55 associated DCIS in the initial core biopsy, representing about half of the samples. In all these cases, the level of STARD3 expression was identical in the invasive and in situ components of the tumors; 49 cases were STARD3-positive and 6 were STARD3-negative (Appendix A).

Collectively, these results show that STARD3 is overexpressed at high levels in the majority of HER2 positive tumors. Its granular staining pattern is consistent with the localization of the protein in endosomes. Interestingly, like HER2, STARD3 overexpression is stable during the transition of in situ to invasive carcinomas.

### 3.4. Immunochemistry on Residual Disease

We conducted an analysis on residual cancer after NST. Among the cohort, we had 56 patients with partial/absent pathological response. Five cases were excluded from the analysis because there was no sufficient material to perform IHC assays (*n* = 4) or because there was a complete response (i.e., no residual breast tumor) on the breast but partial/absent response on the axillary lymph nodes (*n* = 1). Among the 51 patients in whom STARD3 could be evaluated in tumoral resections, we noted that, in most cases (82%, *n* = 43/51), no change for the STARD3 score was observed when comparing initial biopsies with residual disease samples after NST. Three cases had a gain in STARD3 expression (6%, negative to positive, *n* = 3) and 6 cases became negative, with a loss in STARD3 expression (12% positive to negative, *n* = 6) (see Table 3).

First, we investigated whether the change in STARD3 was associated with a change in HER2 expression. We found that when STARD3 score changed after NST, HER2 expression change (loss) was also observed in 67% of cases (versus 2.4% when STARD3 score was unchanged, *p* < 0.001). Second, we investigated whether HER2 amplification detected by SISH was lost after NST. We found that in 4 cases (20%) there was no amplification in the tumoral specimen. These modifications were not associated with STARD3 changes (*p* = 0.403).

Taken together, these data suggest that STARD3 and HER2 expressions remain associated when the HER2 status is lost in the residual disease after NST. This may be explained by the fact that NST may select a tumoral clone which is more resistant to the treatment, or because of initial tumor heterogeneity with the coexistence of two tumors that had been missed by the initial core biopsy.

### 3.5. STARD3 and the Pathological Complete Response

Several anti-HER2 drugs are available in the clinical practice, and the selection of the most effective treatment for each patient is a current challenge. In this context, it is still unclear if determining STARD3 expression together with HER2 expression on the initial biopsy could assist therapy selection. To examine this possibility, we studied the association between STARD3 expression and pCR in our retrospective cohort. We found that a small proportion of HER2-positive breast cancers were STARD3-negative (8/112). However, the majority of them (7/8) fell in the “no pCR” group following neoadjuvant therapy, and the association was significant (Table 2, *p* = 0.028). Tumor heterogeneity regarding STARD3 overexpression in this series was rare (4%, *n* = 3/112 had a positive and a negative contingent), however a higher percentage of STARD3-positive tumor cells was significantly associated with pCR (*p* = 0.015, Table 2). Signal intensity varied between 0, 1+, 2+, and 3+, mean signal intensity was 2.13, and there was no difference between the pCR and “no pCR” groups (Table 2). Regarding staining patterns, the granular staining pattern predominated, out of 112 tumors, 97 had a granular staining pattern while 7 had a diffuse cytoplasmic staining pattern and 8 were negative. The granular pattern was significantly associated with pCR (Table 2). Within the granular staining patterns, we studied the association between either scattered or polarized staining and pCR and we did not find a significant association.

Next, we investigated if these findings were consistent with public databases. We queried the Gene Expression Omnibus (GEO) repository for studies investigating genome expression in HER2-positive breast cancer treated with NST (and similar protocol, see eligibility criteria). We found two studies evaluating gene expression by RNA sequencing, representing *n* = 24 [26] and *n* = 57 specimens [27]. Results are shown in Figure 5. Both distributions suggest that low STARD3 expression is associated with no pCR. However, there was no statistically significant difference between the “no pCR” and “pCR” groups (*p* = 0.18 and *p* = 0.41). Because RNA sequencing does not take into account STARD3 protein localization and function, contrary to IHC, we hypothesize that the number of STARD3-negative samples (weak and diffuse staining) is underestimated. Moreover, the size of the cohorts might be limited to reach statistical power.

Altogether, considering different parameters of STARD3 expression in this series, we found a significant association between the absence of staining, the diffuse staining pattern, and the low number of positive cancer cells with “no PCR” group.

### 3.6. Establishment of a STARD3 Score and Prediction of pCR Using the STARD3 Score

Given that STARD3 is a protein from endosomes and that its function relies on its endosomal localization [32], we thought that a weak diffuse staining could not reflect the presence of an active protein. Based on this reasoning, we built a binary score. Tumors were classified as STARD3-positive (STARD3+) if a granular staining was observed, and as STARD3-negative (STARD3-) if the staining was either absent or diffuse. Then, we evaluated the predictive value of this binary score on pCR. Performances of this score are calculated and reported on in Table 4. Of note, we found that the STARD3 score had a high sensitivity (96%, [88–100%]) and a negative predictive value (87%, [60–98%]), while specificity and positive predictive value were lower (23 and 56%, respectively).

Next, we compared the STARD3 score with other pathological parameters that had been previously associated with pCR. We fitted a logistic regression model, including the STARD3 score, ER and PR negativity, high grade (SBR III), and high Ki67 expression (over 30%), to evaluate pCR versus “no pCR”. Results corresponding to this model are expressed as odd ratio and shown in Table 5. Interestingly, STARD3+ score was significantly associated with pCR with an odd ratio of 5.3 and a 95% confidence interval (CI) 1.24 to 36.44, *p* = 0.044, and this was independent of all other parameters tested in the model. Moreover, in this series, only the STARD3 score was significantly associated with pCR.

Collectively, STARD3 expression quantified with a binary score considering the localization of the protein has a predictive value on NST-treated HER2+ breast cancers.

### 3.7. Prognostic Value of STARD3

We assessed the prognostic value of STARD3 in the entire cohort of HER2-positive breast cancers. For the 112 patients included in the study, the median follow-up period was 71.3 months (range 8.5–185.2). Over the period, 8 deaths occurred (5 related to breast cancer and 3 unrelated) and 16 patients experienced disease relapse (11 distant, 5 local, and 2 both local and distant). Fourteen patients were lost during the follow-up (12.5%) and censored at last contact. Finally, survival outcome was examined in the whole cohort that we separated into two groups based on the STARD3 score. By univariate analysis (Figure 6), comparing survival in the STARD3+ and STARD3− groups, we found no significant difference on overall (*p* = 0.5), breast cancer-specific (*p* = 0.89), local relapse-free (*p* = 0.20), and distant metastasis-free (*p* = 0.7) survivals. Then, we used a multivariate analysis, using a fitted Cox Proportional Hazard model to evaluate STARD3 score on prognosis considering all relevant clinical and pathological parameters: age and stage at diagnosis, hormone receptor expression, pCR, histological subtype, Ki67 expression, SBR grade, and lymphovascular invasion (adjustment parameters). The survival analyses showed that STARD3+ patients did not have significantly different overall (HR 0.15 [95% CI = 0.01, 1.70]; *p* = 0.127), breast cancer-specific (HR = 0.05 [95% CI = 0.00, 2.54], *p* = 0.132), relapse-free (HR 0.63 [95% CI = 0.12, 3.41]; *p* = 0.59), and distant metastasis-free (HR 0.48 [95% CI = 0.08, 2.79]; *p* = 0.41) survivals than STARD3 patients, and this was independent to the adjustment parameters.

Next, we assessed the prognostic value of STARD3 gene expression on all breast cancers using a public online database with KMplot [41]. We computed overall (OS), relapse-free (RFS), and distant metastasis-free (DMFS) survivals comparing groups based on STARD3 (split by upper quartile). These results (reported in Figure 6) showed that high STARD3 expression was associated with shorter OS (hazard ratio [HR] = 1.47; 95% confidence interval [CI] 1.2–1.81; *p* = 0.00022; *n* = 1879 patients), RFS (HR = 1.2; 95% CI 1.07–1.34; *p* = 0.0018; *n* = 4929 patients), and DMFS (HR = 1.3; 95% CI 1.1–1.54; *p* = 0.002; *n* = 2765 patients) survivals. However, after adjustment on HER2, these results showed that STARD3 was not significantly associated with OS (HR = 1.17; 95% CI 0.8–1.71; *p* = 0.4205), RFS (HR = 0.95; 95% CI 0.81–1.12; *p* = 0.5334) and DMFS (HR = 1.03; 95% CI 0.81–1.31; *p* = 0.784) survivals. Conversely, HER2 was significantly associated with RFS and DFMS (respectively *p* = 0.0001 and 0.0039) but not for OS (*p* = 0.1452). Finally, we selected a subset of patients with HER2+ breast cancer and neoadjuvant systemic treatment and found no difference in OS (*p* = 0.4622, *n* = 77 patients), RFS (*p* = 0.4661, *n* = 90 patients), and DFMS (*p* = 0.074, *n* = 49 patients).

Altogether, these data suggest that STARD3 expression is not associated with prognosis in HER2+ breast cancer. Considering all breast cancer subtypes, STARD3 expression is associated with a poorer prognostic, like HER2 expression. The impact of STARD3 on prognosis is not independent from HER2 oncogenic function and is likely a direct consequence of the strong association between HER2 and STARD3 co-amplification and co-expression.

## 4. Discussion

In this study, we assessed the predictive value of STARD3 protein expression on a cohort of primary breast cancer specimens from the HER2-positive subtype that were all treated by neoadjuvant therapy. We evaluated for the first time, to our knowledge, the association of STARD3 expression with pathological complete response (pCR) after NST. We studied the association of STARD3 expression and breast cancer-specific survival and compared it to other relevant patient and tumor characteristics in this HER2-positive series. In addition, we describe a staining and scoring method for STARD3 expression that can be implemented in the routine practice in breast pathology laboratories.

STARD3 is expressed in most tissues at low levels. In addition, STARD3 exists in the whole animal kingdom [42]. Functional studies in cells showed that STARD3 pilots intracellular cholesterol flux from the endoplasmic reticulum to the endosomes/lysosomes [32]. Since STARD3 knock-out mice have no phenotype [43], it is thought that in normal tissues, STARD3 function is balanced by other homeostatic mechanisms regulating cholesterol distribution. Consistent with this idea, silencing STARD3 by siRNAs in HeLa cells (an endometrium HER2-negative cancer cell line) has no impact on viability and cell growth [20]. However, in HER2-amplified cancer, where STARD3 is overexpressed to high levels, functional studies showed that STARD3 silencing is associated with reduced cell growth [19,20,21]. All these studies are consistent and demonstrate that HER2-amplified cancer cells have a specific dependency towards STARD3 expression. It appears that in contrast to other cells, STARD3 function cannot be rescued in HER2-positive cancer cells, indicating a specific role for STARD3 in these cells. Why HER2+ cancer cells are dependent on high STARD3 expression remains unclear, but given its role in cholesterol traffic, it is postulated that STARD3 acts on HER2 cancer cells biology by modulating cholesterol homeostasis. Since STARD3 function on cholesterol rely on its localization in endosomes [32], we reasoned that only the granular staining pattern should reflect the presence of an active protein and should be considered to build an immunohistochemistry score. Of interest, this reasoning is consistent with a previous report on STARD3 protein expression in a Finnish breast cancer patient cohort. The authors suggested that the STARD3 granular (or dot-like) staining pattern should be regarded as positive [35]. In that study, STARD3 staining intensity was graded either negative, low, or high. Samples with high STARD3 expression had a granular staining pattern and were associated with HER2 positivity and reduced survival. Tumors with low expression were not different from negative tumors [35].

### 4.1. STARD3 and Pathological Complete Response

HER2 tumors are heterogeneous and pCR rates are higher compared to other molecular subtypes, underlying the clinical diversity of these tumors [44,45]. pCR rates found in this study are consistent with published studies with anthracycline-based regiments with trastuzumab [8,9,10]. This study suggests that STARD3 expression is correlated with pCR, and therefore that STARD3 may be implicated in anti-HER2 treatment sensitivity and chemo-sensitivity.

Indeed, even though it has been demonstrated that anti-HER2 therapies increase chemo-sensitivity, and despite the fact that combined chemotherapy and anti-HER2 therapy have largely improved HER2-positive breast cancer prognosis, resistances to these treatments exist. De novo and acquired resistances are frequent [12,46,47]. Several biological mechanisms explaining resistance have been implicated, such as signaling from other HER receptors, PI3K/AKT/mTOR activation, MUC4 overexpression, and expression of the p95 isoform of HER2 [48]. Recently, new anti-HER2 targeted therapies have been developed and validated in clinical practice [49], such as conjugated antibody-cytotoxic drugs. For example, trastuzumab-deruxtecan is an HER2-directed antibody-drug conjugate composed of trastuzumab and a cytotoxic topoisomerase I inhibitor [50]. In the phase III Destiny-Breast03 trial, this treatment has shown a significant advantage in terms of progression-free survival compared to T-DM1, another HER2-directed antibody-drug conjugate composed of an antimicrotubule agent attached to trastuzumab [51].

As pCR is an important prognostic factor, STARD3 expression in the initial management of HER2-positive BC could designate some cases with high probability of de novo treatment resistance, therefore providing a rationale to divert from standard medical protocol to increase the probability of pCR and eventually improve patient prognosis. In general, first and next generations of targeted therapies also have significant adverse effects, therefore new predictive indicators of therapeutic response would be useful for patient management.

Of interest, other biomarkers are currently under investigation for predicting pCR. Among them, tumor infiltrating lymphocytes (TIL) have yielded interesting results. A study found that a high proportion of TIL (>50%) in initial biopsies was associated with higher rates of pCR (odds ratio of 6.49; *p* < 0.001) in HER2-positive breast cancer treated with taxanes-cyclophosphamide-trastuzumab-pertuzumab regimens [52]. Another study found that numerous TIL were associated with higher pCR rates in a large cohort [53]. Similarly, a secondary analysis of the NeoALTTO trial (Neoadjuvant Lapatinib and/or Trastuzumab Treatment Optimization) also found an association between TIL and pCR [54]. TIL are implicated in a major hallmark of cancer [55]. We suppose that evaluating STARD3 and TIL could be complementary in pCR prediction in HER2-positive breast cancer.

### 4.2. STARD3 and Prognosis

To date, five clinical studies evaluating STARD3 expression in cancer investigated prognosis [35,36,56,57,58]. Lamy et al. [57] conducted a study including 86 patients with HER2-positive breast cancer (confirmed by in situ hybridization) with a median follow-up of 55 months. They found no association between STARD3 gene amplification (quantified by DNA polymerase chain reaction) with overall and relapse-free survival. Conversely, Vinatzer et al. [36] established a cohort of 85 patients with breast cancer (HER2 negative and positive) with a median follow-up of 10.95 years. They found a worse overall and disease-free survival in case of high STARD3 expression (quantified by RNA reverse-transcriptase polymerase chain reaction), and in patients with HER2 *and* STARD3 high expression, thus suggesting that STARD3 is a bad prognosis marker. However, the authors could not discriminate whether worse outcomes in HER2+/STARD3+ cases were due to HER2 or STARD3 expression, as these two genes are highly correlated. Vassilev et al. published a study including 1193 patients. Among them, 218 had HER2+ breast cancer [35]. They studied STARD3 expression with immunohistochemistry assays and divided their cohorts into three groups based on STARD3 expression: negative (*n* = 538), low (*n* = 538), and high (*n* = 117). Among HER2-positive breast cancers (*n* = 213), 16% were STARD3 negative, a proportion that is similar to this study. The authors found that high STARD3 expression was significantly associated with lower specific survival, however no adjustment was performed on HER2 positivity. Fararjeh et al. [56] conducted a study on an online database (GEPIA, gene expression profile integrative analysis) and found that high levels of STARD3 mRNA were significantly associated with worse overall and event-free survivals. Moreover, they found a significant association between STARD3 mRNA levels and worse overall survival in HER2-positive breast cancers. However, no data were shown on event-free and specific survivals among this subgroup. Finally, Li et al. recently published a study on 1641 primary breast cancers cases [58]. STARD3 expression was evaluated with immunohistochemistry assay. They found 16.6% of STARD3-positive tumors, and STARD3 expression was strongly correlated to HER2 expression. This team also found that patients with STARD3+ breast cancers had significantly lower overall survival, but no adjustment was performed on HER2 positivity.

In this study, we found that on the contrary HER2+/STARD3+ patients with NST had better overall survival compared to HER2+/STARD3− ones. This discrepancy can be explained by (1) the fact that included populations were different as this study included only HER2-positive breast cancers with NST; and (2) STARD3 was measured differently (protein expression or gene amplification/expression). In addition, this study investigated STARD3 prognostic value independently of HER2 and other significant clinical and pathological characteristics.

### 4.3. STARD3 as a Therapeutic Target

Given the in vitro dependence of HER2-positive cells to STARD3, and the findings of this study, STARD3 can be a possible therapeutic target in HER2-positive breast cancer. Indeed, a STARD3 inhibitor was recently developed and tested by Lapillo et al. [59] on different cancer cell lines, including breast and colon. In the context of heterogeneity of HER2-positive breast cancers, and the frequent resistance mechanisms developed against anti-HER2 targeted therapies, a novel therapeutic target such as STARD3 may become a good alternative.

Furthermore, STARD3 could also be implicated in other HER2-positive cancers, such as ovarian, gastric, pulmonary, vesical, and prostatic cancers. Finally, STARD3 has a potential interest in other non-tumoral pathologies, such as hepatitis C and atherosclerosis. RNA interference-mediated knockdowns identified STARD3 as necessary for Hepatitis C Virus replication, as it depends on endosomal cholesterol homeostasis [60]. Additionally, it has been shown that STARD3 protein over-expression in dysregulated macrophages restores an impaired cholesterol homeostasis and induces an anti-atherogenic macrophage lipid phenotype, positing a potentially therapeutic strategy [61].

### 4.4. Limitations

This work has some limits. Indeed, this study was conducted in a retrospective setting, and this can lead to inclusion biases. To eliminate this possibility, we read HER2 staining in all cases, and we performed in situ hybridization in the case of equivocal results. As a result, seven cases were excluded after a second reading by the pathologist and were classified as HER2-negative, and because of the loss of contact with some patients during follow-up. Moreover, HER2-positivity was determined by measuring protein expression (IHC) and not HER2 gene amplification (in situ hybridization). Finally, the amplification status remains unknown in the majority of cases. In this cohort, we included only anthracycline-based chemotherapy, as it is the standard of care in our hospital. Therefore, these results cannot be extrapolated to other regimens. In addition, the performance of the STARD3 score resulted in a low specificity (i.e., high false positive rate). However, as false positives correspond to patients with positive STARD3 score which do not achieve pCR, they will receive standard of care treatment. On the contrary, the high sensitivity (i.e., low false negative rate) remains the most important parameter as it lowers potential risks of iatrogeny (i.e., changing NST protocol for patient who would not have achieved pCR with standard of care). Nonetheless, further investigations are needed to improve specificity, possibly with other clinical and pathological parameters. Finally, few events were recorded during the follow-up, thus limiting interpretation of the results; a longer follow-up is needed to confirm the impact of STARD3 expression on prognosis.

## 5. Conclusions

In the light of these findings, we believe it is of prime interest to explore the potential of STARD3 protein as a predictive marker for treatment response. STARD3 expression is able to identify a subgroup of HER2-positive BC with a high risk of no pCR and provides a rational to investigate a distinct neoadjuvant systemic regimen in this subgroup. Future studies are needed on independent cohorts and larger numbers in a prospective setting to evaluate the impact of STARD3 expression on pCR and on prognosis.

## Figures and Tables

**Figure 1 cancers-15-00362-f001:**
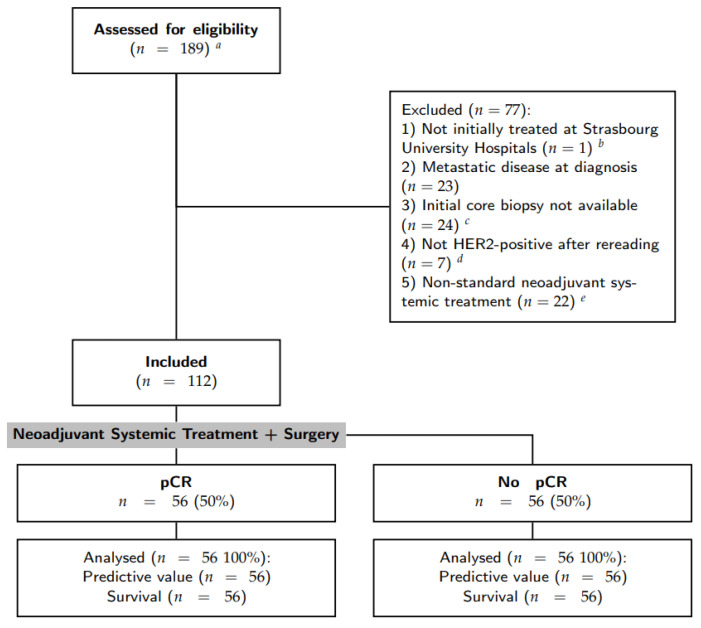
Flow chart of patient selection. *^a^* The initial query was all breast specimens HER2+ treated by surgery; *^b^* one patient was excluded as it was a second pathology opinion after a surgery in another center; *^c^* some initial core biopsies were stored in other pathology laboratories. Requests for stored material in other laboratories were made 3 times, some were recovered but others were not available or did not have residual material to perform immunohistochemistry assays; *^d^* either HER2 score 0–1 or 2+ with non-amplified in situ hybridization; *^e^* without anthracyclines and/or with pertuzumab.

**Figure 2 cancers-15-00362-f002:**
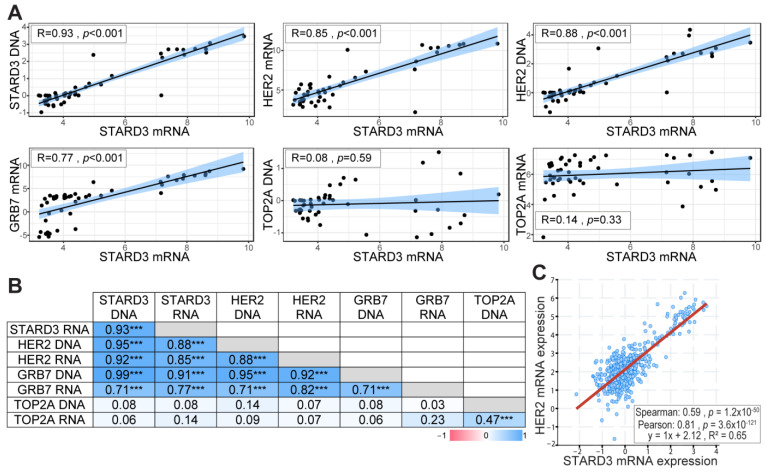
STARD3 and HER2 expression are correlated in breast cancers. (**A**) Pearson correlation coefficient and regression line showing the correlation between STARD3 DNA, HER2 RNA, HER2 DNA, GRB7 RNA, TOP2A DNA, TOP2A RNA, and STARD3 RNA. RNA and DNA data were obtained from the Cancer Cell Line Encyclopedia database. RNA expressions are Log2 values, DNA represents copy number variation. (**B**) Pearson correlation coefficient (r) matrix and its statistical significance (*** = *p* < 0.001) between STARD3, HER2, GRB7, TOP2A DNA, and RNA. STARD3 is correlated with HER2 and GRB7 but not with TOP2A. (**C**) Correlation between STARD3 and HER2 mRNA expression (microarray, log2 values) in a cohort of breast cancers from The Cancer Genome Atlas (TCGA) database (Cancer Genome Atlas Network, 2012).

**Figure 3 cancers-15-00362-f003:**
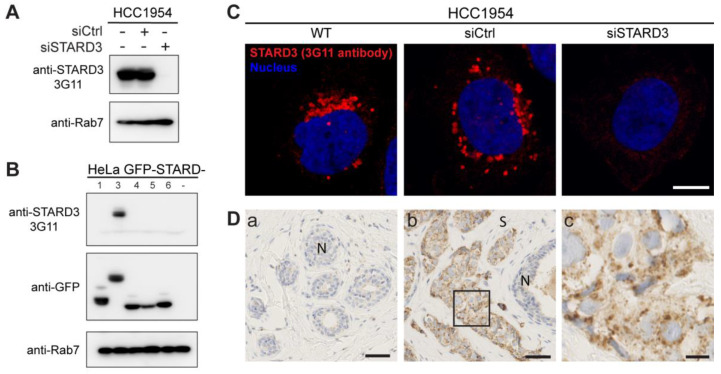
Generation of a specific anti-STARD3 antibody for immunohistochemistry. (**A**) Western blot analysis of STARD3 expression using the anti-STARD3 antibody 3G11 in HCC1954 cells (WT and transfected with control (siCtrl) and STARD3-targeting (siSTARD3)-siRNAs). Anti-Rab7 antibody was used as a loading control. (**B**) Western blot analysis of HeLa cells transiently expressing STARD1, STARD3, STARD4, STARD5, and STARD6 fused to GFP (GFP-STARD) using anti-STARD3 3G11 antibody. Anti-Rab7 antibody was used as a loading control. (**C**) HCC1954 cells (WT) and HCC1954 cells transfected with control siRNAs (siCtrl), and siRNAs targeting STARD3 (siSTARD3) were labeled with anti-STARD3 3G11 antibodies (red). Nuclei are stained blue. Scale bar: 10 μm. (**D**) Immunohistochemistry on normal breast (**a**) and HER2+ breast tumor (**b**) sections using anti-STARD3 3G11 antibody. N marks the position of a normal duct. S marks the tumor stroma. The subpanel (**c**) is a higher magnification image (4.5×) of the area outlined in black in (**b**). Scale bar: 50 μm (**a**,**b**) and 10 μm (**c**). All the whole western blot figures can be found in Appendix A.

**Figure 4 cancers-15-00362-f004:**
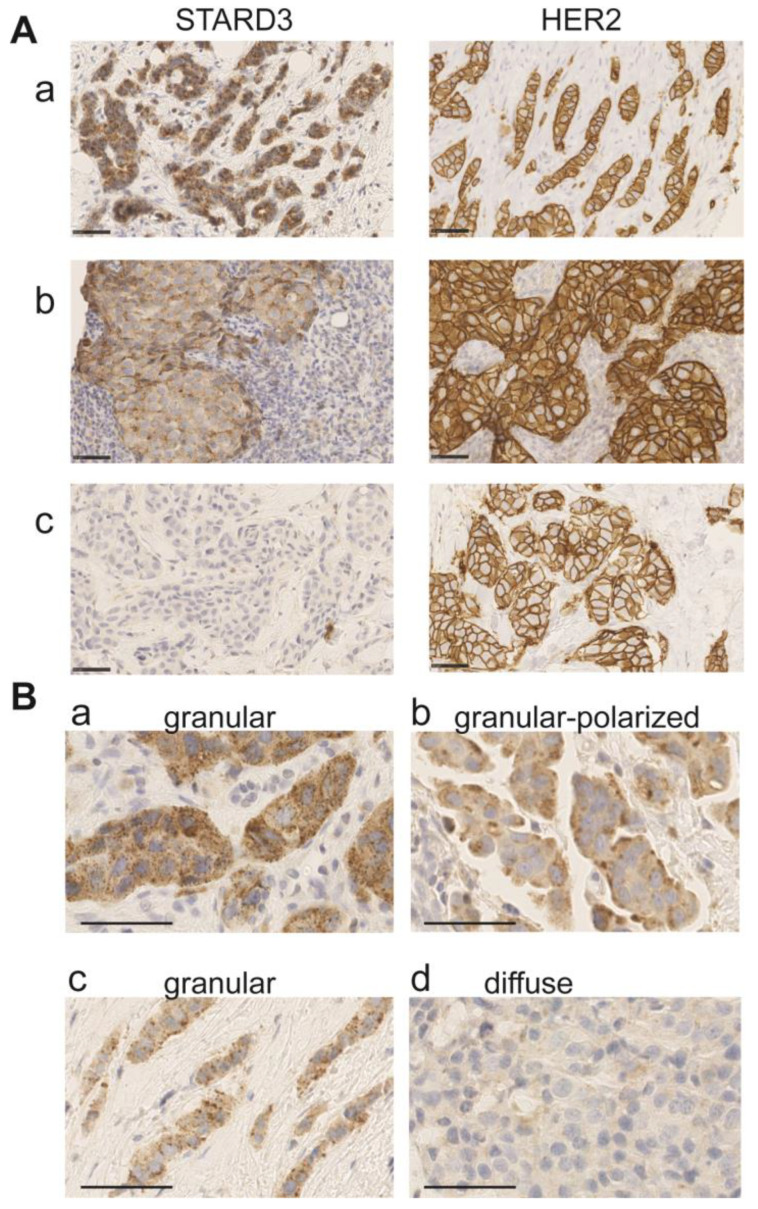
STARD3 staining patterns in breast cancer. (**A**) Representative immunohistochemistry signals for STARD3 and HER2 on three biopsies. STARD3 protein is expressed at high levels in the majority of HER2-positive tumors (**a**,**b**). STARD3 expression is absent in some HER2-positive tumors (**c**). (**B**) Staining patterns for STARD3 in breast tumors. STARD3 is predominantly detected with a granular scattered pattern in the cytoplasm of cancer cells (**a**,**c**). In some cases, the granular STARD3 staining is polarized (**b**). In a few cases, the staining is weak and diffuse in the cytoplasm (**d**). Bars are 50 μm, nuclei are counterstained in blue with hematoxylin.

**Figure 5 cancers-15-00362-f005:**
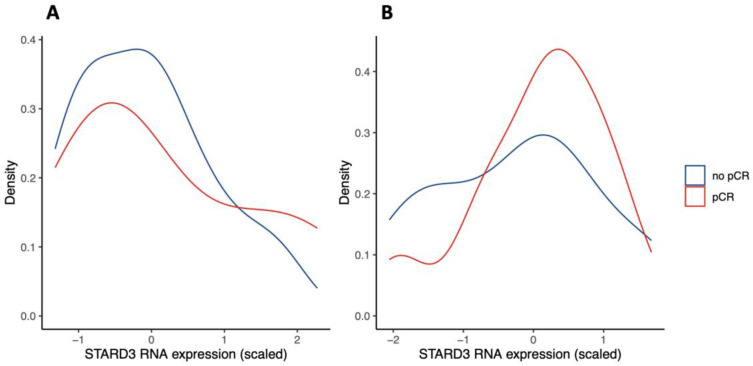
STARD3 gene expression in Gene Expression Omnibus (GEO) repository studies (density plots). Legend: panel (**A**) shows scaled STARD3 gene expression (RNA sequencing) in a study of *n* = 24 initial biopsies [26]. Panel (**B**) shows scaled STARD3 gene expression (RNA sequencing) in a study of *n* = 57 initial biopsies [27].

**Figure 6 cancers-15-00362-f006:**
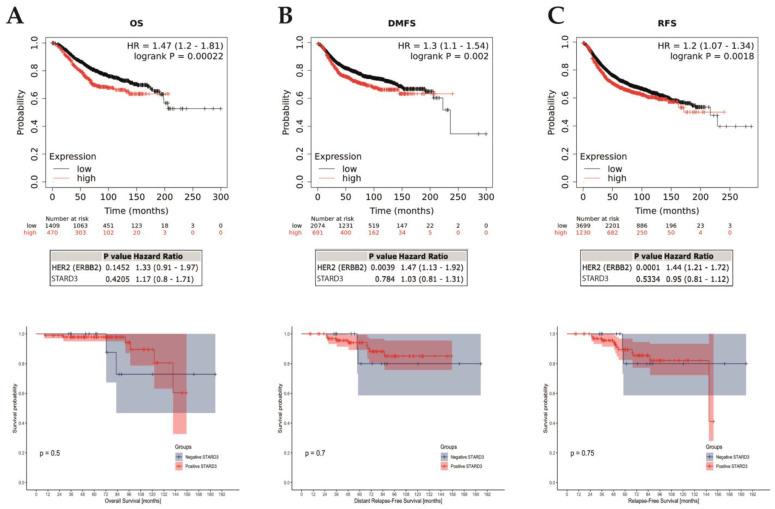
STARD3 prognosis. Association of STARD3 mRNA expression with survival (**A**): overall [OS]; (**B**): distant-metastasis free [DFMS]; (**C**): relapse-free [RFS]) in a cohort of breast cancers from the Kaplan–Meier plotter (KM plotter database) in the upper row, with multivariate analysis on HER2 below. The last row shows survivals in the studied cohort. Of interest, there are cross-overs in the survival curves, which indicate a non-proportional hazard. Consequently, in this survival analysis, *p*-values (log-rank tests) must be interpreted with caution, given the greater risk of wrongly concluding a statistically significant difference.

**Table 1 cancers-15-00362-t001:** Cohort description according to pCR.

	pCR N = 56	No pCR N = 56	Total N = 112	*p*-Value
Age				
Mean (SD)	53 (11.4)	54 (12.0)	54 (11.7)	0.735
Range	34–76	27–85	27–85	
Multifocal disease				
Yes	9 (16.1%)	10 (17.9%)	19 (17.0%)	0.801
No	47 (83.9%)	46 (82.1%)	93 (83.0%)	
Histological subtype				
Ductal	53 (94.6%)	54 (96.4%)	107 (95.5%)	0.310
Lobular	1 (1.8%)	2 (3.6%)	3 (2.7%)	
Lymphoïd stroma	2 (3.6%)	0 (0%)	2 (1.8%)	
Initial size (mm)				
Mean (SD)	34.8 (21.1)	37.2 (24.4)	36.0 (22.7)	0.588
Range	5–110	10–140	5–140	
Initial node involvement (TNM classification)				
cN0	24 (42.9%)	27 (48.2%)	51 (45.5%)	0.481
cN1	27 (48.2%)	27 (48.2%)	54 (48.2%)	
cN2	5 (8.9%)	2 (3.6%)	7 (6.2%)	
ER H-score				
Mean (SD)	91.8 (114.9)	178.8 (118.4)	135.2 (124.1)	<0.001
Range	0–300	0–300	0–300	
RP H-score				
Mean (SD)	50.3 (95.5)	75.5 (107.0)	62.9 (101.8)	0.191
Range	0–300	0–300	0–300	
High grade (SBR III)				
No	15 (26.8%)	29 (51.8%)	44 (39.3%)	0.007
Yes	41 (73.2%)	27 (48.2%)	68 (60.7%)	
High Ki67 (≥30%)				
Unknown	2	1	3	0.052
No	9 (16.7%)	18 (32.7%)	27 (24.8%)	
Yes	45 (83.3%)	37 (67.3%)	82 (75.2%)	
NST protocol				
EC-TH	18 (32.1%)	20 (35.7%)	38 (33.9%)	0.285
FEC-TH	34 (60.7%)	32 (57.2%)	66 (58.9%)	
TCH	0 (0%)	2 (3.6%)	2 (1.8%)	
TH	4 (7.1%)	2 (3.6%)	6 (5.4%)	
Breast surgery				
Radical mastectomy	22 (39.3%)	19 (33.9%)	41 (36.6%)	0.556
Conservative treatment	34 (60.7%)	37 (66.1%)	71 (63.4%)	
Breast surgery				
Sentinel lymph node biopsy	5 (8.9%)	11 (19.6%)	16 (14.3%)	0.105
Axillary lymph node dissection	51 (91.1%)	45 (80.4%)	96 (85.7%)	

Legend: pCR = pathological complete response; SD = standard deviation; ER = estrogen receptor; PR = progesterone receptors; SBR = Scarff–Bloom–Richardson; NST = neoadjuvant systemic treatment; EC-TH = epirubicin-cyclophospamide taxane-trastuzumab; FEC-TH fluorouracil-epirubicin-cyclophospamide taxane-trastuzumab; TCH = taxane-cyclophosphamide-trastuzumab; TH = taxane -trastuzumab; STARD3 = StAR-related lipid transfer domain-3.

**Table 2 cancers-15-00362-t002:** STARD3 immunohistochemistry assay according to pCR.

	pCR N = 56	No pCR N = 56	Total N = 112	*p*-Value
STARD3 negative signal				
No	55 (98.2%)	49 (87.5%)	104 (92.9%)	0.028
Yes	1 (1.8%)	7 (12.5%)	8 (7.1%)	
STARD3 intensity				
Mean (SD)	2.28 (0.80)	1.99 (0.98)	2.13 (0.90)	0.093
Range	0–3	0–3	0–3	
STARD3 distribution				
Absent	1 (1.8%)	7 (12.5%)	8 (7.1%)	0.088
Diffuse	49 (87.5%)	44 (78.6%)	93 (83.0%)	
Marginal	6 (10.7%)	5 (8.9%)	11 (9.8%)	
STARD3 granular pattern				
Not granular	3 (5.4%)	12 (21.4%)	15 (13.4%)	0.013
Granular	53 (94.6%)	44 (78.6%)	97 (86.6%)	
STARD3 cell positivity (%)				
Mean (SD)	97 (16.6)	84 (36.2)	90 (28.8)	0.015
Range	0–100	0–100	0–100	

**Table 3 cancers-15-00362-t003:** STARD3 and HER2 expression after neoadjuvant systemic treatment.

STARD3 Score			HER2	Post-NST ISH		
Initial	Post-NST	Change	Post-NST	Amplification	No Amplification	Detail
Negative	Negative	No (*n* = 8)	Negative (*n* = 1)	0	1	No change (*n* = 1)
			Positive (*n* = 7)	7	0	No change (*n* = 6) Change: loss of amplification (*n* = 1)
	Positive	Yes: gain (*n* = 3)	Negative (*n* = 1)	0	1	No change (*n* = 1)
			Positive (*n* = 2)	2	0	No change (*n* = 2)
Positive	Negative	Yes: loss (*n* = 6)	Negative (*n* = 5)	3	2	No change (*n* = 3) Change: loss of amplification (*n* = 2)
			Positive (*n* = 1)	1	0	No change (*n* = 1)
	Positive	No (*n* = 34)	Positive	Not performed (*n* = 31)		

Legend: pCR = pathological complete response; NST = neoadjuvant systemic treatment; STARD3 = StAR-related lipid transfer domain-3; HER2 = Human epidermal growth-factor receptor 2; SISH = in situ hybridization.

**Table 4 cancers-15-00362-t004:** STARD3 score performances on pCR prediction.

Characteristics	Value (95% CI)
True prevalence	0.50 (0.40, 0.60)
Sensitivity	0.96 (0.88, 1.00)
Specificity	0.23 (0.13, 0.36)
Positive predictive value	0.56 (0.45, 0.66)
Negative predictive value	0.87 (0.60, 0.98)
Correctly classified proportion	0.60 (0.50, 0.69)

Legend: CI = confidence interval.

**Table 5 cancers-15-00362-t005:** Multivariate analysis of pathological parameters impact on pCR.

Parameter	Odds Ratio	2.5%	97.5%	*p*-Value
STARD3 granular signal	5.24	1.24	36.42	0.044 *
ER negativity	2.70	0.94	8.15	0.069
PR negativity	1.20	0.43	3.27	0.729
SBR III	2.05	0.81	5.26	0.129
Ki67 ≥ 30%	1.29	0.44	3.80	0.641

Legend: STARD3 = StAR-related lipid transfer domain-3; ER = estrogen receptor; PR = progesterone receptor; SBR = Scarff–Bloom–Richardson. * *p* < 0.05

## Data Availability

The data presented in this study are available on request from the corresponding author.

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
