# Peer review of "STARD3: A New Biomarker in HER2-Positive Breast Cancer"

_cancers, 2023, doi:10.3390/cancers15020362_

Round 1

Reviewer 1 Report

It is a well-written manuscript in which the authors analyze the association between STAD3 and pathological complete response and survival in HER2-positive breast cancer. Several questions need to be answered.

1. Validation cohort is recommended to validate the main findings.

2. Specificity is quite low for STAD3 score performances in pCR prediction, which needs further solution.

3. Table 4 and Table 5 have the same title.

4. Survival curves in Figure 5 show crossover, which could affect the results of STARD3 prognosis.

Author Response

Dear Reviewer, 

We would like to thank you for your high-quality reviews and comments. We think that it has improved the overall quality of the manuscript as it pointed out issues we did not see initially. In addition to the correction suggested by the reviewers, we added some minor modifications (mainly typos).

Please find below the point-by-point responses to your comments (in italic)

  1. Validation cohort is recommended to validate the main findings.

--> we agree on the fact that supplementary data are necessary to validate these findings. Data on the value of STARD3 expression on prognostic were published by independent studies from other teams their conclusions are consistent and we cite and discuss them (see discussion). On the other hand, this is the first study investigating STARD3 protein expression on pCR, and is the result of several years of work. In fact, recruitment took over ten years, the replication of this study with another cohort would be ideal but is not feasible in the provided time. Moreover, with the publication of this study, we will contact and hope to be contacted by other teams interested to conduct a similar study on their cohort. Nonetheless, we wanted to add new data to this manuscript, because we agree with the issue you pointed out (and future readers). Consequently, we gathered data on STARD3 expression and pCR from GEO database and found 2 studies with gene expression profiles (RNAseq) with the similar eligibility criteria. We added these new results in the manuscript, along with a new figure (Figure 5). 

  1. Specificity is quite low for STARD3 score performances in pCR prediction, which needs further solution.

--> as for any test, one must advise whether it is best to have high sensitivity or high specificity. Within this context, we reasoned that it is mainly the negative predictive value of this test that would be clinically relevant (i.e. the absence of pCR if the STARD3 score is negative). Indeed, low specificity means a higher false positivity rate (STARD3+ with no pCR), which does not impact patient prognosis as they receive the standard of care. On the contrary, the high sensitivity means that the false negative rate (STARD3- which achieve pCR) is low. If in those patients an eventual change of NST protocol will be discussed, the aim of the test would be to have less patients which could have achieved pCR without those changes. Still, future investigations are needed to improve specificity, possibly with other clinical and pathological features. For these reasons, we privileged specificity over sensitivity. We added some explanations in the discussion as this point was not clear (in subsection 4.4. Limitations).

  1. Table 4 and Table 5 have the same title.

--> we corrected the Table 5 title with " Multivariate analysis of pathological parameters impact on pCR"

  1. Survival curves in Figure 5 show crossover, which could affect the results of STARD3 prognosis.

--> crossover indicates non proportional hazards. This is generally a result of the survival times having greater variance in one group than another. The risk in this situation is mainly type II, which means the non-rejection of the null hypothesis which ought have been [1]. This was considered in the interpretation of the results, and should not interfere with the manuscript hypothesis. Still, we added some explanations in the manuscript to underline it (Figure 6 legend)

  1. Bouliotis, G.; Billingham, L. Crossing survival curves: alternatives to the log-rank test. Trials 2011, 12, A137, doi:10.1186/1745-6215-12-S1-A137.

Reviewer 2 Report

This manuscript by Tomasetto and co-workers is an interesting investigation. They studied the possible role of stard3 in her-2 positive breast cancer and strongly supported the hypothesis that this protein is involved and its level of expression could be considered as a biomarker.

The opinion of this referee is to accept the manuscript as it is.

Author Response

Dear Reviewer, we thank you for your positive comment

Reviewer 3 Report

It is an interesting study which may be of worth to be published.

I would recommend that the authors could or should add other markers or pathology or clinical parameters for pCR and compare with STARD3.One of the example is TILs( tumor infiltrating lymphocytes) or other findings in the biopsy could be evaluated together with HER2 expression.  Also add other clinical parameters 

Author Response

Dear Reviewer, 

We would like to thank you for your high-quality review and comments. We think that it has improved the overall quality of the manuscript as you pointed out issues we did not see initially.

Please find below the responses to your comments:

--> TILs : we added a paragraph in the discussion about TILs which we think could be complementary to STARD3 in pCR prediction. As TILs are not part of the routine assessment today in our center, we did not evaluate it on initial biopsies and cannot therefore analyze it and its implications with STARD3. However, it would be interesting to conduct a prospective trial including TILs. 

--> we added other clinical parameters in table 1 however they did not impact the STARD3 score performances.

Round 2

Reviewer 1 Report

The manuscript has been sufficiently improved to be accepted.